# Effect of Phytohormones Supplementation under Nitrogen Depletion on Biomass and Lipid Production of *Nannochloropsis oceanica* for Integrated Application in Nutrition and Biodiesel

**Hussein El-Sayed Touliabah** [1,2] **and Adel W. Almutairi** [1,*]

[1] Biological Sciences Department, Rabigh-Faculty of Science and Arts, King Abdulaziz University, Rabigh 21911, Saudi Arabia; hehassan@kau.edu.sa

[2] Botany Department, Faculty of women for Arts, Science & Education, Ain Shams University, Cairo 11757, Egypt

[*] Correspondence: aalmutairi@kau.edu.sa; Tel.: +966-56-993-4096

**Abstract:** Economic viability of biodiesel production relies mainly on the productivity of biomass and microalgal lipids. In addition, production of omega fatty acids is favorable for human nutrition. Thus, enhancement of lipid accumulation with high proportion of omega fatty acids could help the dual use of microalgal lipids in human nutrition and biodiesel production through biorefinery. In that context, phytohormones have been identified as a promising factor to increase biomass and lipids production. However, nitrogen limitation has been discussed as a potential tool for lipid accumulation in microalgae, which results in simultaneous growth retardation. The present study aims to investigate the combined effect of N-depletion and 3-Indoleacetic acid (IAA) supplementation on lipid accumulation of the marine eustigmatophyte *Nannochloropsis oceanica* as one of the promising microalgae for omega fatty acids production. The study confirmed that N-starvation stimulates the lipid content of *N. oceanica*. IAA enhanced both growth and lipid accumulation due to enhancement of pigments biosynthesis. Therefore, combination effect of IAA and nitrogen depletion showed gradual increase in the dry weight compared to the control. Lipid analysis showed lower quantity of saturated fatty acids (SFA, 26.25%) than the sum of monounsaturated (MUFA) and polyunsaturated fatty acids (PUFA). Under N-depletion, SFA decreased by 12.98% compared to the control, which recorded much reduction by increasing of IAA concentration. Reduction of SFA was in favor of PUFA, mainly omega-6 and omega-3 fatty acids which increased significantly due to IAA combined with N-depletion. Thus, the present study suggests a biorefinery approach for lipids extracted from *N. oceanica* for dual application in nutrition followed by biodiesel production.

**Keywords:** integrated impact; phytohormones; biofuels; renewable energy; *Nannochloropsis oceanica*; essential fatty acids

## 1. Introduction

The increased global use of energy led to rising of oil prices and reduction in fossil fuel reservoirs, with concurrent environmental and human health issues due to harmful emissions. These issues have prompted researchers to look for new energy sources that are sustainable, eco-friendly, and renewable [1,2]. Nowadays, there is consensus that microalgae lipids are a renewable alternative feedstock for biofuel production that has many advantages over other feedstocks. Microalgae-derived lipids provide a safe substitute for petroleum diesel due to their sustainability, biodegradability, non-toxicity, and low greenhouse gas (GHG) emissions [3]. There are several advantages for using microalgae as a third-generation biofuel over the first- and second-generations which include food

crops and non-food wastes, respectively. For example, microalgae are photosynthetic organisms with high growth rates (estimated by about 20–30 fold higher than oil crops), can grow in non-arable lands using wastewater or seawater, and can be harvested during the whole year with relatively high lipids productivity [4,5].

Microalgal cells are an optimum natural supplier for neutral lipids as a potential reserve of fuel, characterized by rapid cell division and high photosynthetic efficiency [6]. Application of different stress conditions, particularly nutrient starvation, leads to lipid bodies accumulation in the form of triacylglycerides (TAG). Among different nutrients, phytohormones are considered as a category of growth-accelerating chemicals found naturally in plants and algae at very low concentrations, and can be divided into several groups such as auxins, cytokinins, gibberellins, brassinosteroids, jasmonates and strigolactones [7]. These phytohormones are important for abiotic stress-tolerance in many crops [8], and affect the metabolic processes, cell division, propagation, elderliness, nutrient utilization, and sometimes inhibit the cellular metabolism. Therefore, these compounds in low concentrations can control the algal cellular activities and biochemical composition [9].

Microalgal capacity to alter the lipid biosynthesis under unfavorable conditions has contributed to various lipid augmentation strategies such as salt stress and nutrient deprivation [3,10,11]. In that context, nitrogen (N) stress is considered beneficial because of its easy implementation and strong impact on lipid content. Most nitrogen salts used for microalgae cultivation are ammonium salts, which can easily be converted to amino acids [2,12–15]. However, due to the metabolic pathway shifted to storage lipid biosynthesis, nitrogen depletion usually leads to reduced cell division and biomass production [16]. Converti et al. [17] evaluated the lipid yield of *Nannochloropsis oculata* and *Chlorella vulgaris* under nitrogen limitation. They recorded the significant increase of lipid yield by 15.31% and 16.41%, respectively, in both species under nitrogen-limited conditions. Similar results were obtained by Gao et al. [18] where nitrogen limitation increased lipid accumulation from 23% to 46%, while biomass productivity decreased from 19 mg $L^{-1}$ $d^{-1}$ to 12 mg $L^{-1}$ $d^{-1}$ in *Chaetoceros muelleri*. Oleaginous algae can alter their lipid-biosynthesis pathways to shape and accumulate neutral lipids as TAG. In some oleaginous algae, TAG increased by over 50% of the total dry biomass due to alteration of growth conditions [19,20]. Pal et al. [21] reported that the impact of abiotic factors such as salinity on nitrogen repletion contributes to neutral lipid production in *Nannochloropsis* sp. In addition, Jeyakumar et al. [22] investigated the relationship between the high nitrogen concentration and the changes in PUFA of *Isochrysis* sp. They concluded that high nitrogen concentrations enhance biomass production, with lower lipid content. However, nitrogen starvation enhanced lipid accumulation and specifically the eicosapentaenoic acid (EPA) synthesis in *Isochrysis* sp. The latter is an omega-3 fatty acid has excellent anti-inflammatory properties and important for prevention of neurodegenerative disorders, cognitive function growth and cardiovascular diseases.

The genus *Nannochloropsis* has six known species which can live in varied ranges of salinity [23]. *Nannochloropsis oceanica* is considered a viable commercial alga for processing high polyunsaturated fatty acids [24] and was discussed recently as a promising candidate for biodiesel production [25]. For the best of authors' knowledge, the combined effect of N-depletion and 3-Indoleacetic acid (IAA) supplementation on fatty acid profile of *N. oceanica* was not previously evaluated. Therefore, the present study aimed to explore the changes in the growth, pigments, lipids, and fatty acids of *N. oceanica* under five different concentrations of IAA with/without N-depletion. Fatty acid methyl esters (FAMEs) were synthesized from the extracted lipids, and biodiesel properties were evaluated.

## 2. Materials and Methods

### 2.1. Microalgal Strain and Growth

The phytohormone IAA was purchased from Sigma Aldrich (no. I2886). *N. oceanica* (CCAP 849/10) was supplied from the Culture Collection of Algae and Protozoa (CCAP), United Kingdom. The alga was cultivated in 1000 mL Erlenmeyer flasks with 500 mL of f/2 medium as described by CCAP according to Guillard and Ryther [26]. Exponentially grown culture was used to provide the inoculum, and the new cultures were inoculated at initial optical density ($OD_{650}$) of 0.05. After inoculation, cultures were incubated at 25 ± 1 °C with a light emitting diodes (LEDs) illumination at light intensity of 100 $\mu$mol m$^{-2}$ s$^{-1}$, and 12:12-h light: dark cycle.

### 2.2. Experimental Design

A total of 12 treatments were performed using different concentrations of the growth hormone and/or nitrogen concentrations. The treatments included nitrogen depletion, IAA supplementation (0.1, 1.0, 2.5, 5.0 and 10.0 ppm), and combined effect of N-depletion and IAA supplementation. A control culture was grown using standard f/2 medium. The impact of the applied treatments on growth, chlorophyll contents, lipid content and fatty acid profile were studied.

### 2.3. Growth Determination

From a preliminary experiment by daily monitoring the culture optical density at 650 nm, *N. oceanica* grown under N-stress showed the end of exponential phase at day 12, which was defined as the harvest time for further analysis. Biomass productivity of *N. oceanica* was calculated as the dry weight at the last day of experiment divided by the incubation time [27]. Briefly, 20 mL of algal culture were centrifuged at 4000 rpm for 15 min, the supernatant was removed, and the pellet was freeze-dried at −80 °C. The weight of the dried biomass was determined gravimetrically.

### 2.4. Chlorophyll Content, Fatty Acid Profile, and FAMEs Characteristics

Pigments of *N. oceanica* were extracted as described by Roy [28]. Total lipids were estimated according to Folch et al. [29]. Fatty acid profile was determined in the extracted lipids as fatty acid methyl esters (FAMEs) according to the method defined by Abomohra et al. [30]. Then, the converted FAMEs were identified using gas chromatography mass spectroscopy (GC-MS). The main FAMEs characteristics at different treatments were evaluated based on fatty acid profile according to Hoekman [31], as previously described by Abomohra et al. [32] using the following equations;

$$ADU = \sum n \times M_f \tag{1}$$

$$KV = -0.6313\,ADU + 5.2065 \tag{2}$$

$$SG = 0.0055\,ADU + 0.8726 \tag{3}$$

$$CP = -13.356\,ADU + 19.994 \tag{4}$$

$$CN = -6.6684\,ADU + 62.876 \tag{5}$$

$$IV = 74.373\,ADU + 12.71 \tag{6}$$

$$HHV = 1.7601\,ADU + 38.534 \tag{7}$$

where the main characteristics included the average unsaturation degree (*ADU*), specific gravity (*SG*), cetane number (*CN*), kinematic viscosity (*KV*), cloud point (*CP*), iodine value (*IV*), and higher heating value (*HHV*). *n* is the number of C=C double bonds each fatty acid with a mass fraction of $M_f$.

### 2.5. Statistical Analysis

One-way analysis of variance (ANOVA) was conducted for three replicates using SPSS program Ver. 20. Data are represented as the mean of the replicates ± standard deviation (SD). The differences were considered low and high statistically significant at probability $p < 0.05$ and 0.01, respectively.

## 3. Results and Discussion

### 3.1. Effect on Biomass Production

The present study examined the effect of IAA and nitrogen depletion, separately or in combination, on biomass production and lipid content of *N. oceanica* (Figure 1). The dry weight varied significantly according to the different combinations of nitrogen depletion and IAA concentrations. It fluctuated between 123.6–382.3 mg $L^{-1}$, showing the lowest value at T2 (Nitrogen depletion) and the highest value at T5 (single effect of 2.5 mg $L^{-1}$ IAA). The reduction of dry weight at T2 was 41.7% lower than the control (T1). The present data agreed with those obtained by Merzlyak et al. [33], who stated that 20% reduction in the dry weight was recorded under nitrogen malnutrition. In the case of hormone supplementation, the present study showed a gradual increase in growth by increasing of IAA up to T5 (which represented 129.0% higher than the control), then showed gradual reduction at higher IAA concentrations. The present results agree with the data obtained by Fabian et al. [34] who reported that the treatment of *Chlorella sorokiniana* with 10 mg $L^{-1}$ IAA results in the highest dry weight, representing about 9-folds higher than the control. A similar effect was recorded with *Parachlorella kessleri*, where a positive correlation was recorded with IAA and microalgal growth [35]. Like the effect of high concentrations of IAA obtained in the present study, Udayen et al. [7] stated that the high concentrations of methyl jasmonate (MeJA) and salicylic acid (SA) showed inhibitory effects on the growth of *N. oceanica*. On the other hand, combination of hormone supplementation with nitrogen depletion (T8–T12) showed a gradual increase of the dry weight by increasing of IAA but was still lower than the control. These results confirmed that IAA has a significant impact on the growth (F value = 17.51, $p < 0.001$), which mitigated the negative effect of N-depletion. Biomass productivity showed a similar trend, where T5 and T6 showed the highest values of 31.9 and 30.9 mg $L^{-1}$ $d^{-1}$, respectively (Figure 2).

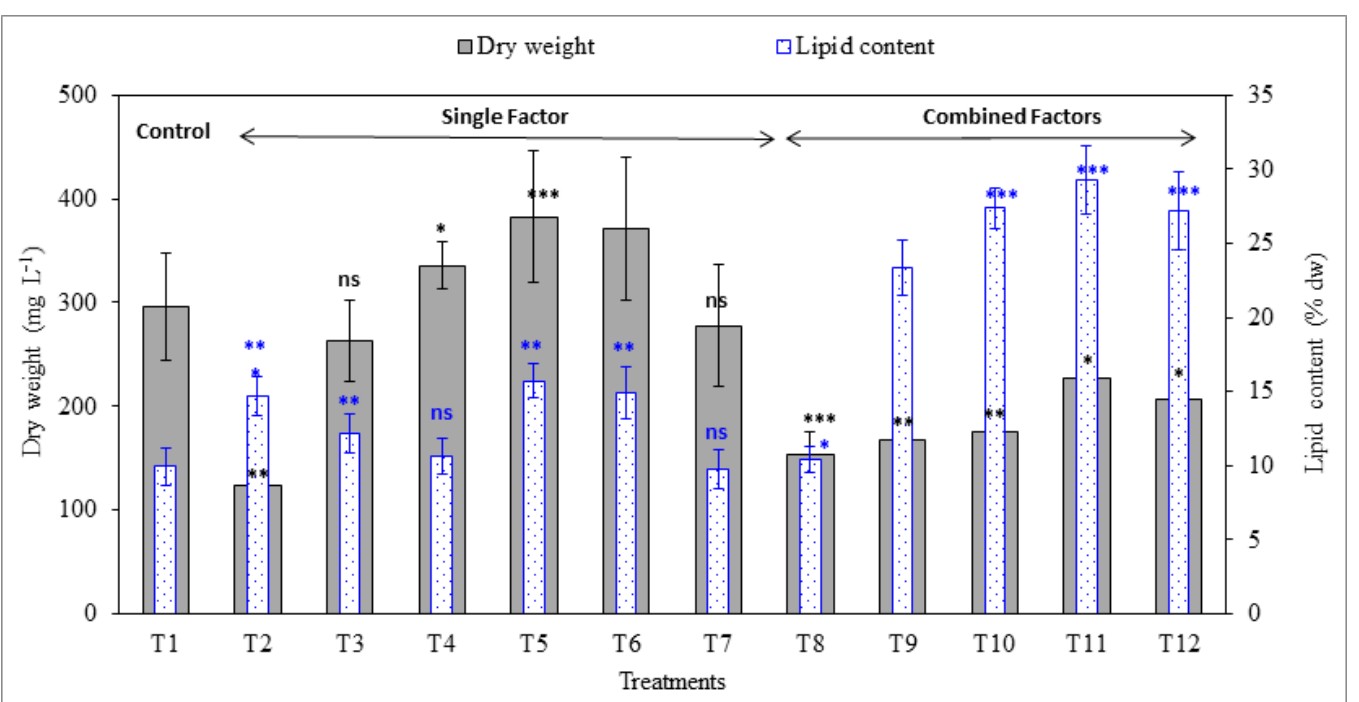

**Figure 1.** Effect of N-depletion and/or indole acetic acid supplementation on the dry weight and lipid content of *Nanno-chloropsis oceanica* grown for 12 days in modified f/2 medium. All tests were performed in triplicates (*n* = 3) and the error bars represent the SD. * *p* < 0.05, ** *p* < 0.01, *** *p* < 0.001 with respect to the control. T1 = Control, T2 = N-depletion, T3 = 0.1 ppm IAA, T4 = 1.0 ppm IAA. T5 = 2.5 ppm IAA, T6 = 5.0 ppm IAA, T7 = 10.0 ppm IAA, T8 = N-depletion + 0.1 ppm IAA, T9 = N-depletion + 1.0 ppm IAA, T10 = N-depletion + 2.5 ppm IAA, T11 = N-depletion + 5.0 ppm IAA, T12 = N-depletion + 10.0 ppm IAA.

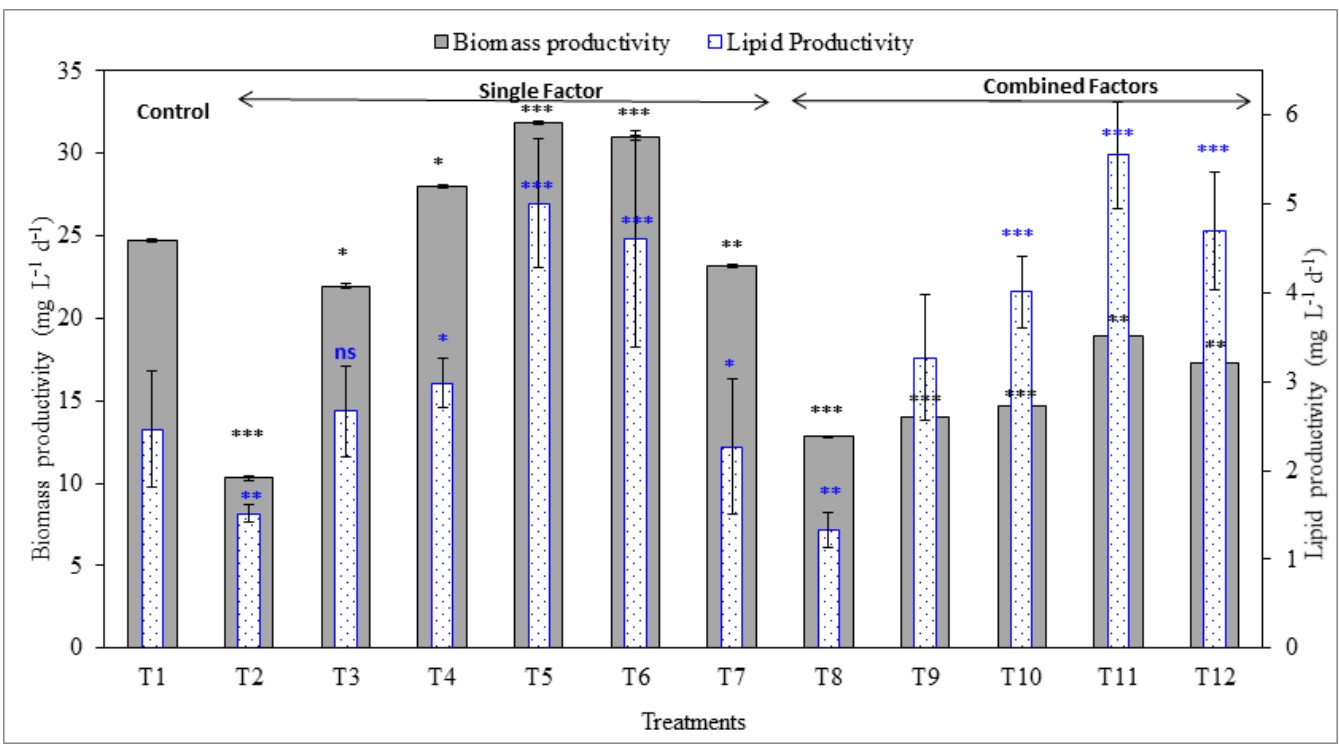

**Figure 2.** Effect of N-depletion and/or indole acetic acid supplementation on biomass and lipid productivities of *Nanno-chloropsis oceanica* grown for 12 days in modified f/2 medium. All tests were performed in triplicates (*n* = 3) and the error bars represent the SD. * *p* < 0.05, ** *p* < 0.01, *** *p* < 0.001 with respect to the control. T1 = Control, T2 = N-depletion, T3 = 0.1 ppm IAA, T4 = 1.0 ppm IAA. T5 = 2.5 ppm IAA, T6 = 5.0 ppm IAA, T7 = 10.0 ppm IAA, T8 = N-depletion + 0.1 ppm IAA, T9 = N-depletion + 1.0 ppm IAA, T10 = N-depletion + 2.5 ppm IAA, T11 = N-depletion + 5.0 ppm IAA, T12 = N-depletion + 10.0 ppm IAA.

### 3.2. Effect on the Lipid Accumulation

It is widely documented that nitrogen starvation stimulates lipid accumulation in the algal cells. The N-depleted culture (T2) showed enhancement of lipid content by about 1.5-fold (147.3%) compared to the control (Figure 1). Results also indicated a very powerful changes in the lipid content between the different treatments and the control (F value = 44.67, *p* < 0.001). Similarly, Converti et al. [17] reported that lipid yield increased by 15.3% and 16.4% in *Nannochloropsis oculata* and *C. vulgaris*, respectively, under nitrogen starvation. Moussa et al. [36] found that the lipid accumulation increased due to biosynthesis of cellular lipids during cultivation of *Tetraselmis marina*. Various stressors such as temperature, light intensity, nitrogen, ionic radiation, or phosphorus starvation were reported to increase the lipid accumulation in microalgae [21,37]. Interestingly, IAA treatment showed an increase in lipid content by increasing the hormone concentration up to T6 (which showed 149.4% increase over the control). Thus, combined treatments of N-depletion and IAA showed significant increase in lipid content compared to N-depletion or IAA separately, where T11 showed increment by about 3 folds compared to the control. Due to reduction of lipid content at T7, combined N-depletion with 10 ppm IAA resulted in reduction in lipid content by 1.8% lower than T11 (Figure 1). According to Converti et al. [17], lipid content in *Chlorella vulgaris* and *Nannochloropsis oculate* increased three-fold

with decreasing the nitrogen concentration in the medium, and phytohormones was suggested as a hopeful way to enhance lipid and biomass productivity of microalgae under unfavorable conditions of the nitrogen starvation. Due to enhancement of biomass production and lipid accumulation in T11 (N-depletion + 5.0 ppm IAA), it showed the highest lipid productivity of 5.6 mg L$^{-1}$ d$^{-1}$, which represented 1.5-time and 98.2% higher than the control and T7 (10.0 ppm IAA), respectively (Figure 2). Thus, from quantitative aspect, combined effect of N-depletion and supplementation of 5.0 ppm IAA is suggested to stimulate lipid productivity.

### 3.3. Effect on Pigments

Since the nutrition stress is reflected in microalgal cells by the changes in Chl-a content, the effect of IAA and/or N-depletion on the Chl-a content was monitored (Figure 3). Chl-a decreased under nitrogen depletion, which resulted in reduction of biomass as discussed in the previous section. The chlorophyll reduction in T2 was estimated by 73.34% below the control (T1). However, treatment with different concentrations of IAA up to 1.0 ppm (T4) led to enhancement of Chl-a by 189.60% higher than the control, then showed gradual reduction at higher IAA concentrations. On the other hand, the increasing of hormone concentration under nitrogen starvation resulted in increase in the Chl-a content up to T11 (about two-fold higher than the control), then a sharp decline was recorded in T12, reaching 68.52% of the control. It is noticeable that Chl-a content exhibited the same dry weight pattern. It can be explained by the close relationship between growth and photosynthetic activity which his controlled mainly by Chl-a content. The present study showed decomposition of Chl-a which was correlated to the reduction in the dry weight (*r* = 0.68). Providing hormones as carbon source and growth stimulants enhanced the chlorophyll content and cellular growth, in addition to mitigating the negative impact of nitrogen limitation.

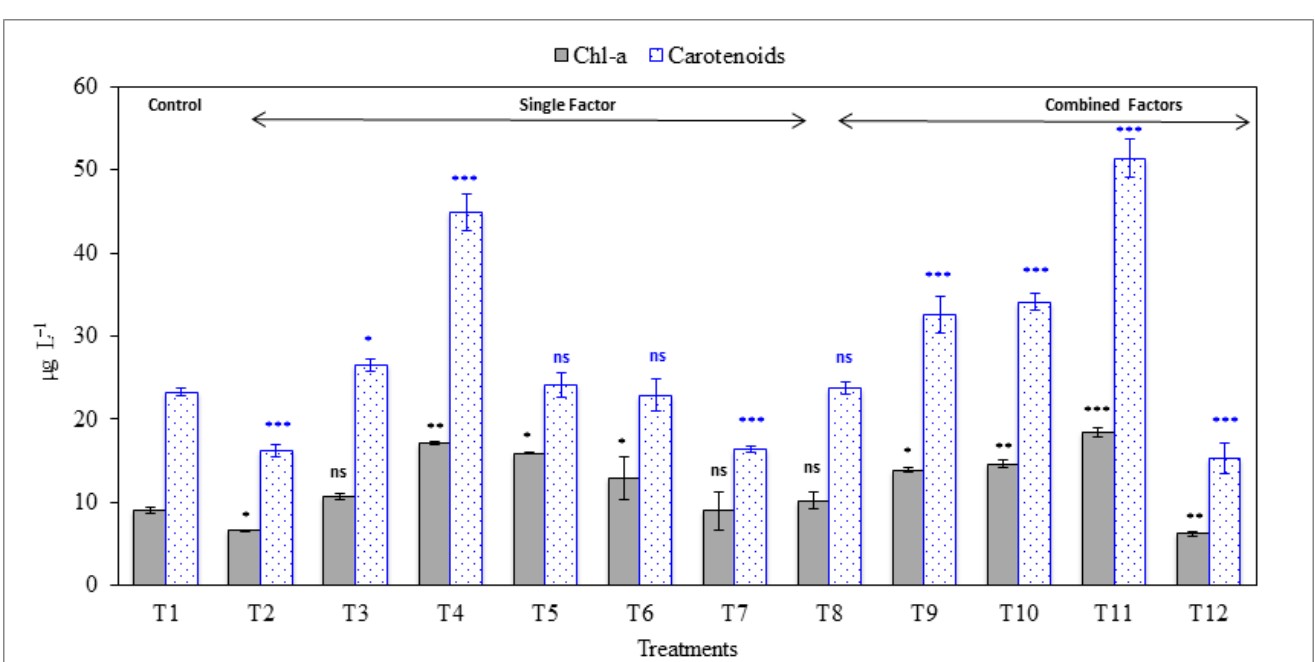

**Figure 3.** Effect of N-depletion and/or indole acetic acid supplementation on the chlorophyll-a and carotenoids of *Nannochloropsis oceanica* grown for 12 days in modified f/2 medium. All tests were performed in triplicates (*n* = 3) and the error bars represent the SD. * *p* < 0.05, ** *p* < 0.01, *** *p* < 0.001 with respect to the control. T1 = Control, T2 = N-depletion, T3 = 0.1 ppm IAA, T4 = 1.0 ppm IAA. T5 = 2.5 ppm IAA, T6 = 5.0 ppm IAA, T7 = 10.0 ppm IAA, T8 = N-depletion + 0.1 ppm IAA, T9 = N-depletion + 1.0 ppm IAA, T10 = N-depletion + 2.5 ppm IAA, T11 = N-depletion + 5.0 ppm IAA, T12 = N-depletion + 10.0 ppm IAA.

Carotenoids showed the same pattern as Chl-a due to nitrogen deficiency (Figure 3). This resulted in a significant shortage of carotenoids by up to 69.72% (T2) in comparison to the control. On the other hand, a significant increase in carotenoids was recorded by increasing of IAA, where the maximum value of carotenoids was shown in T4, representing 1.9-fold higher than the control. However, further increase in IAA resulted in significant reduction of carotenoids, reaching 70.6% in T7 compared to the control. Under the combined effect of IAA and N-limitation, there was a rise in carotenoids by 2.2-fold in T11 compared to the control, followed by significant reduction by 65.51% in T12 (Figure 3).

*3.4. Effects on Fatty Acids Profile and Biodiesel Properties.*

Fatty acid profile of *N. oceanic* showed lower proportion of saturated fatty acids (SFA, 26.3%) than unsaturated fatty acids (73.8%) (Figure 4). The unsaturated fatty acids comprised monounsaturated (MUFA, 28.9%) and polyunsaturated fatty acids (PUFA, 44.8%). Most of PUFA we omega family fatty acids (Table 1), which play an important role in the human diet and physiology. The present results showed lower SFA content than the oleaginous *N. oceanica* isolated from Eastern Harbor of Alexandria Coast, Egypt and grown in f/2 medium, which showed 69.1% SFA [25]. This finding confirms that fatty acid profile varies in different isolates within the same species and significantly affected by the growth conditions. Thus, optimization of growth parameters and/or medium is essential to obtain a favorable fatty acid profile for biodiesel production.

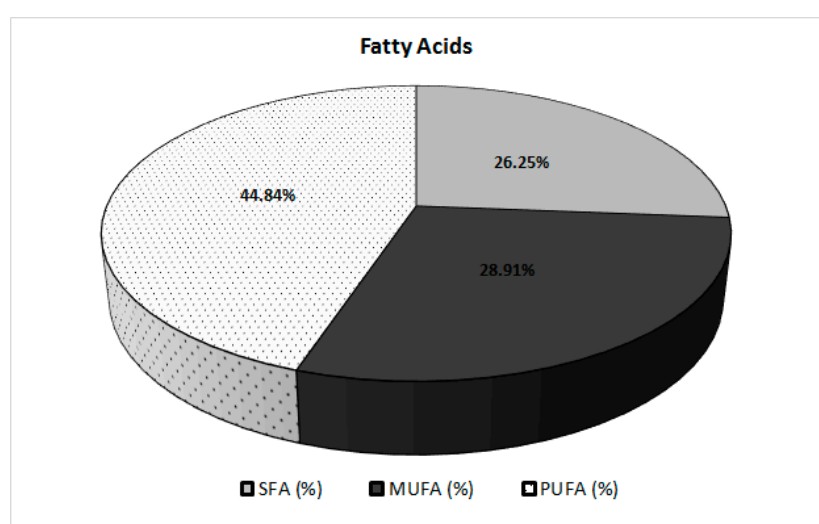

**Figure 4.** The percentage of saturated and unsaturated fatty acids of *Nannochloropsis oceanic* grown in f/2 medium for 12 days.

Under the N-depletion (T2), the SFA decreased by 12.98% compared to the control (T1). In addition, the SFA significantly decreased by increasing of IAA up to T7, where the it decreased by about 35.77% with respect to the control (Figure 5). Thus, the dual effect of N-depletion and IAA resulted in sharp reduction in SFA, with the maximum reduction in T11 (by 67.83% lower than the control). PUFA showed similar trend where N-depletion resulted in reduction of PUFA by 9.40% lower than the control. However, IAA supplementation up to T4 increased PUFA by 36.11% compared to the control, then showed further increase at higher concentrations, reaching 1.5-fold at T9. Reduction of SFA and PUFA under N-depletion was in favor of MUFA, which increased in T2 by 7.42% over the control. Similar to PUFA, IAA and combined effect of N-depletion with IAA to T11 enhanced MFUA by 30.93% comparing to the control (Figure 5).

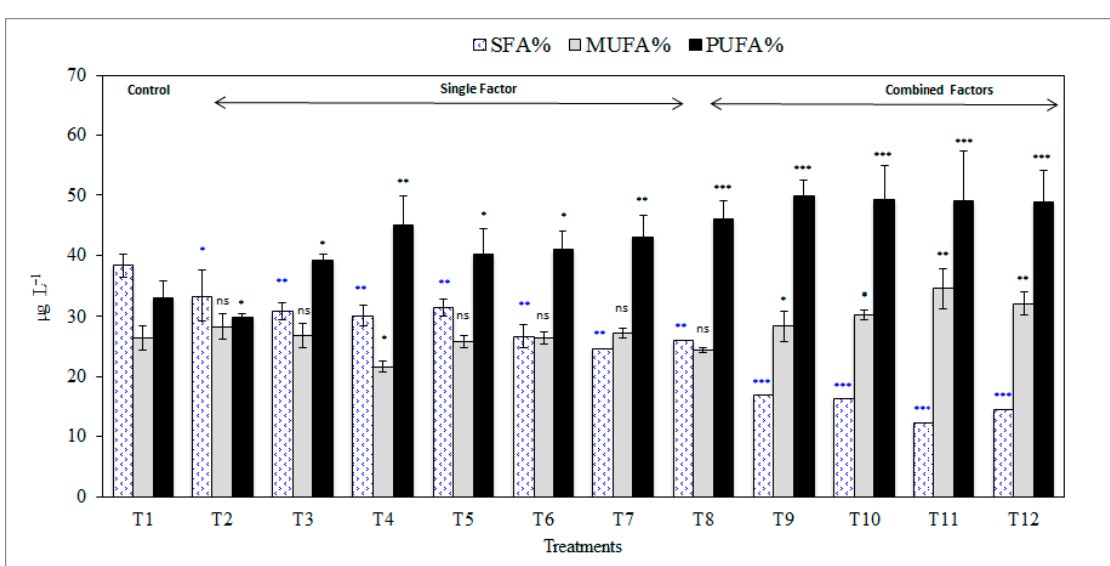

**Figure 5.** Effect of N-depletion and/or indole acetic acid supplementation on fatty acid groups of *Nannochloropsis oceanica* grown for 12 days in modified f/2 medium. All tests were performed in triplicates (*n* = 3) and the error bars represent the SD. * *p* < 0.05, ** *p* < 0.01, *** *p* < 0.001 with respect to the control. T1 = Control, T2 = N-depletion, T3 = 0.1 ppm IAA, T4 = 1.0 ppm IAA. T5 = 2.5 ppm IAA, T6 = 5.0 ppm IAA, T7 = 10.0 ppm IAA, T8 = N-depletion + 0.1 ppm IAA, T9 = N-depletion + 1.0 ppm IAA, T10 = N-depletion + 2.5 ppm IAA, T11 = N-depletion + 5.0 ppm IAA, T12 = N-depletion + 10.0 ppm IAA.

Extensive research studies have reported that several microalgae species can accumulate high amount of lipids and oils under environmental stresses [38]. In addition, it has been thoroughly documented that phytohormones mediate a variety of microalgal activities to ameliorate the environmental stresses [39]. In addition, phytohormones, including IAA, work interactively to enhance cellular biosynthesis of a variety of useful molecules, especially lipids [40–42]. Enhancement of microalgal biomass with high-value products using phytohormone can facilitate the scale-up microalgae cultivation for marketable production of useful products and commodities. Amongst, many studies have shown that IAA maintained outstanding impact on fatty acid profile of microalgae. Accordingly, investigating the effects of plant hormones, including IAA, on lipid profile of microalgae seems a proper method for enhanced biodiesel production.

Concerning biodiesel properties, results showed high degree of unsaturation due to the treatment with N-depletion and/or IAA supplementation (Table 2). This resulted in increasing of iodine value higher than the recommended values of ≤120 g I₂/100 g oil [43,44], with reduction of cetane number from 53.45 in the control to 49.49 in T12. On the other hand, the HHV increased by increasing the IAA concentration with or without N-depletion, reaching the maximum value of 42.07, which represented 2.5% higher than the control. Increasing of HHV is advantageous as it represents the energy content in a certain amount of oil. However, the stability of the oil is expected to decrease by increasing the IAA concentration due to the high degree of unsaturation in FAMEs. Thus, pretreatment of oil for separation of unsaturated fatty acids for nutritional applications could serve dual purpose of enhancing the process economy and the biodiesel characteristics from the residual oil.

Overall, it is confirmed that studying and defining the possible impact of plant hormones on growth and metabolic activities of microalgae maintain a great potential for "Microalgae-Green Revolution" which will enhance the renewable production of useful biochemicals, alleviating the global warming and to combat the eminent global energy crisis [45]. Because IAA supplementation resulted in enhancement of PUFAs which include omega family fatty acids, the present study suggests biorefinery of lipids from microalgae for first separation of essential fatty acids for feed purpose, while use the residual lipids for biodiesel production. However, such concept needs further validation and economic evaluation, which might enhance the economic feasibility of microalgal biodiesel production in the future.

**Table 1.** Fatty acid profile of *Nannochloropsis oceanica* grown for 12 days in modified f/2 medium with different treatments of N-depletion and/or indole acetic acid supplementation. T1 = Control, T2 = N-depletion, T3 = 0.1 ppm IAA, T4 = 1.0 ppm IAA. T5 = 2.5 ppm IAA, T6 = 5.0 ppm IAA, T7 = 10.0 ppm IAA, T8 = N-depletion + 0.1 ppm IAA, T9 = N-depletion + 1.0 ppm IAA, T10 = N-depletion + 2.5 ppm IAA, T11 = N-depletion + 5.0 ppm IAA, T12 = N-depletion + 10.0 ppm IAA.

| Fatty Acids | T1 | T2 | T3 | T4 | T5 | T6 | T7 | T8 | T9 | T10 | T11 | T12 |
|---|---|---|---|---|---|---|---|---|---|---|---|---|
| Butyric Acid (C4:0) | 3.09 ± 0.09 | 3.63 ± 0.11 | 3.01 ± 0.19 | 2.61 ± 0.26 | 4.46 ± 0.31 | 3.14 ± 0.27 | 3.66 ± 0.45 | 3.08 ± 0.31 | 2.13 ± 0.24 | 3.01 ± 0.40 | 2.04 ± 0.24 | 3.19 ± 0.20 |
| Lauric Acid (C12:0) | 7.19 ± 0.52 | 5.46 ± 0.29 | 6.19 ± 0.48 | 5.43 ± 0.55 | 4.98 ± 0.68 | 5.87 ± 0.67 | 6.12 ± 0.86 | 4.92 ± 0.65 | 3.18 ± 0.22 | 4.14 ± 0.43 | 3.87 ± 0.13 | 2.58 ± 0.42 |
| Myristic (14:0) | 2.48 ± 0.40 | 2.44 ± 0.16 | 2.19 ± 0.20 | 3.18 ± 0.32 | 3.68 ± 0.32 | 3.07 ± 0.07 | 2.01 ± 0.35 | 2.94 ± 0.62 | 2.41 ± 0.37 | 2.04 ± 0.21 | 2.00 ± 0.38 | 3.45 ± 0.45 |
| Palmitic (16:0) | 12.08 ± 1.02 | 10.28 ± 0.01 | 9.13 ± 0.68 | 8.02 ± 0.20 | 8.61 ± 0.70 | 5.42 ± 0.64 | 5.14 ± 0.98 | 6.21 ± 0.93 | 4.19 ± 0.33 | 3.15 ± 0.36 | 1.18 ± 0.19 | 2.07 ± 0.26 |
| Stearic (18:0) | 5.19 ± 1.04 | 4.29 ± 0.64 | 2.45 ± 0.44 | 3.82 ± 0.41 | 2.52 ± 0.47 | 3.58 ± 0.53 | 2.09 ± 0.22 | 2.66 ± 0.01 | 0.95 ± 0.19 | 0.84 ± 0.14 | 0.87 ± 0.16 | 1.04 ± 0.15 |
| Arachidilic (C20:0) | 8.29 ± 0.96 | 7.28 ± 0.73 | 7.91 ± 1.02 | 6.97 ± 1.09 | 7.21 ± 0.93 | 6.58 ± 0.33 | 6.61 ± 0.49 | 6.27 ± 0.98 | 3.98 ± 0.79 | 3.14 ± 0.40 | 2.27 ± 0.38 | 2.18 ± 0.35 |
| Myristoleic (14:1) | 3.89 ± 0.41 | 2.89 ± 0.50 | 2.45 ± 0.31 | 2.51 ± 0.33 | 4.08 ± 0.10 | 4.28 ± 0.59 | 5.26 ± 0.44 | 3.17 ± 0.38 | 4.97 ± 0.89 | 4.96 ± 0.26 | 3.97 ± 0.58 | 4.07 ± 0.21 |
| Palmitoleic (16:1) | 6.09 ± 0.98 | 6.13 ± 0.77 | 8.08 ± 0.95 | 6.42 ± 0.94 | 7.19 ± 1.11 | 7.11 ± 1.16 | 6.27 ± 0.78 | 4.63 ± 0.37 | 7.16 ± 0.75 | 8.74 ± 1.12 | 8.56 ± 0.98 | 8.22 ± 0.83 |
| Oleic (18:1) | 15.39 ± 1.84 | 19.31 ± 1.74 | 16.33 ± 2.06 | 12.65 ± 0.57 | 14.55 ± 0.75 | 15.09 ± 0.93 | 15.63 ± 0.89 | 16.52 ± 0.75 | 16.21 ± 0.51 | 16.54 ± 1.59 | 21.00 ± 0.97 | 19.81 ± 1.26 |
| Linoleic (18:2; ω-6) | 12.56 ± 1.13 | 16.08 ± 1.04 | 16.07 ± 0.22 | 24.09 ± 0.34 | 16.27 ± 0.60 | 15.22 ± 0.70 | 16.24 ± 0.75 | 16.01 ± 1.26 | 20.24 ± 0.99 | 19.65 ± 0.67 | 20.81 ± 0.89 | 19.88 ± 1.43 |
| α-Linolenic (18:3; ω-3) | 4.36 ± 0.90 | 4.62 ± 1.11 | 5.95 ± 0.95 | 6.28 ± 0.93 | 8.41 ± 1.26 | 6.46 ± 1.47 | 7.14 ± 0.90 | 9.42 ± 0.65 | 8.19 ± 0.96 | 8.43 ± 0.40 | 6.91 ± 0.91 | 8.09 ± 0.31 |
| γ-Linolenic (18:3; ω-6) | 5.28 ± 0.64 | 6.15 ± 1.06 | 5.81 ± 0.01 | 5.81 ± 1.82 | 6.19 ± 0.77 | 7.99 ± 1.08 | 8.07 ± 0.01 | 10.37 ± 1.20 | 9.23 ± 0.69 | 10.16 ± 1.74 | 9.16 ± 1.19 | 8.34 ± 1.02 |
| Arachidonic (20:4) | 7.19 ± 0.84 | 6.12 ± 0.40 | 7.63 ± 0.69 | 5.92 ± 0.75 | 6.28 ± 1.05 | 8.41 ± 0.64 | 9.48 ± 0.66 | 6.23 ± 0.74 | 9.15 ± 0.85 | 8.24 ± 1.76 | 9.17 ± 0.77 | 9.08 ± 0.47 |
| Eicosapentaenoic (20:5; ω-3) | 3.66 ± 0.67 | 2.82 ± 0.30 | 3.65 ± 0.79 | 2.89 ± 0.52 | 3.08 ± 0.61 | 2.98 ± 0.33 | 2.18 ± 0.67 | 4.08 ± 0.43 | 3.17 ± 0.54 | 2.91 ± 0.46 | 3.08 ± 0.27 | 3.87 ± 0.23 |
| Docosahexanoic (22:6; ω-3) | 2.49 ± 0.32 | 2.03 ± 0.27 | 2.94 ± 0.11 | 2.65 ± 0.57 | 2.44 ± 0.49 | 4.89 ± 0.46 | 4.91 ± 0.62 | 3.18 ± 0.32 | 4.62 ± 0.50 | 3.87 ± 0.17 | 4.89 ± 0.25 | 3.98 ± 0.57 |

**Table 2.** Biodiesel properties of lipid extracted from *Nannochloropsis oceanica* grown for 12 days in modified f/2 medium with different treatments of N-depletion and/or indole acetic acid supplementation. T1 = Control, T2 = N-depletion, T3 = 0.1 ppm IAA, T4 = 1.0 ppm IAA. T5 = 2.5 ppm IAA, T6 = 5.0 ppm IAA, T7 = 10.0 ppm IAA, T8 = N-depletion + 0.1 ppm IAA, T9 = N-depletion + 1.0 ppm IAA, T10 = N-depletion + 2.5 ppm IAA, T11 = N-depletion + 5.0 ppm IAA, T12 = N-depletion + 10.0 ppm IAA.

| Properties | T1 | T2 | T3 | T4 | T5 | T6 | T7 | T8 | T9 | T10 | T11 | T12 | US | EN |
|---|---|---|---|---|---|---|---|---|---|---|---|---|---|---|
| *ADU* | 1.41 | 1.44 | 1.61 | 1.60 | 1.57 | 1.78 | 1.84 | 1.80 | 2.01 | 1.96 | 2.05 | 2.01 | - | - |
| *KV* | 4.31 | 4.30 | 4.19 | 4.20 | 4.21 | 4.08 | 4.05 | 4.07 | 3.94 | 3.97 | 3.91 | 3.94 | 1.9–6.0 | 3.5–5.0 |
| *SG* | 0.88 | 0.88 | 0.88 | 0.88 | 0.88 | 0.88 | 0.88 | 0.88 | 0.88 | 0.88 | 0.88 | 0.88 | 0.85–0.9 | - |
| *CP* | 1.11 | 0.82 | −1.47 | −1.38 | −1.02 | −3.80 | −4.52 | −4.06 | −6.88 | −6.19 | −7.36 | −6.81 | - | - |
| *CN* | 53.45 | 53.30 | 52.16 | 52.20 | 52.39 | 51.00 | 50.64 | 50.87 | 49.46 | 49.80 | 49.22 | 49.49 | Min. 47 | 51–120 |
| *IV* | 117.86 | 119.48 | 132.22 | 131.75 | 129.71 | 145.21 | 149.22 | 146.66 | 162.39 | 158.51 | 165.01 | 161.98 | - | Max. 120 |
| *HHV* | 41.02 | 41.06 | 41.36 | 41.35 | 41.30 | 41.67 | 41.76 | 41.70 | 42.08 | 41.98 | 42.14 | 42.07 | - | - |

*ADU* Degree of unsaturation; *KV* Kinematic viscosity; *SG* Specific gravity; *CP* Cloud point; *CN* Cetane number; *IV* Iodine value; *HHV* Higher heating value; US American international standards (ASTM D6751-08, 2008), and EN European international standards (EN 14214, 2008).

## 4. Conclusions

The present study provides a great insight into lipid accumulation in one of the marine oleaginous microalgae under N-depletion combined with IAA supplementation for enhanced biodiesel production and nutritional value by stimulating of lipid accumulation and omega fatty acids, respectively. Results showed enhancement of growth and lipid accumulation under the combined effect, which significantly enhanced the lipid productivity up to 5.6 mg L$^{-1}$ d$^{-1}$. The present study suggests a novel approach of possible dual use of lipid for nutrition due to presence of high proportion of omega fatty acids, followed by biodiesel production from the lipid residue through a biorefinery approach. Such an approach could enhance the economy of the process and the biodiesel characteristics. Future research is required to evaluate the suggested approach of dual use of lipids extracted from *N. oceanica* under IAA supplementation combined with N-stress. In addition, changes in cell number, cellular volume, lipid content and fatty acid profile over the incubation time at the optimized conditions could provide a good understanding to the mechanism of lipid accumulation.

**Author Contributions:** Conceptualization, H.E.S.-T. and A.W.A.; methodology, H.E.S.-T. and A.W.A.; formal analysis, H.E.S.-T. and A.W.A.; investigation, H.E.S.-T. and A.W.A.; writing—original draft preparation, H.E.S.-T. and A.W.A.; writing—review and editing, H.E.S.-T. and A.W.A.; funding acquisition, H.E.S.-T. and A.W.A. All authors have read and agreed to the published version of the manuscript.

**Funding:** This work was funded by the Deanship of Scientific Research (DSR), King Abdulaziz University (Jeddah, Saudi Arabia) by grant No. G: 232/662/1440.

**Acknowledgments:** The authors, therefore, gratefully acknowledge the DSR for technical and financial support.

**Conflicts of Interest:** The authors declare no conflict of interest.

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
