# Peer review of "Effect of Phytohormones Supplementation under Nitrogen Depletion on Biomass and Lipid Production of Nannochloropsis oceanica for Integrated Application in Nutrition and Biodiesel"

_sustainability, doi:10.3390/su13020592_

Round 1

Reviewer 1 Report

Please list and define T1-T12 in each figure legend. It is very inconvenient to search for each when looking at figures and interpreting results.

Line 55 phytochromes are light receptive molecules...I believe the preferred term should be phytohormones

The statement that IAA "will mitigate the impact of N-depletion on growth" is not clearly defined or intuitive. Cells cannot continue to grow without a source of nitrogen for amino acids and protein synthesis. The observed increases in cell dry weight seem to be lipids accumulating (from the authors data). No cell numbers are given so difficult to distinguish growth from accumulation of products.

The experiment was conducted over 12 days. This is a prolonged nitrogen stress and often results in cell death and degradation. The lipid and biomass data were acquired at the end of this period. It would be much more impactful and informative if the data were collected throughout (daily) to understand the rates and timing of lipid accumulation impacted by IAA. This would also define the optimal point for harvesting to achieve maximal sustainable yields.

Does the rate change over time as cells increase in stress? Do cell number increases account for some of the biomass increase (ie using the IAA as a nitrogen source and/ or hormonal regulation)?

This work demonstrates a significant impact for increasing lipids that can be harvested with minimal resource inputs (nitrogen and subsequent biomass). Defining the functional parameters that optimize these yields is critical. The amount of biomass that is derived from lipids vs cell material can be calculated from this data. It would help with interpretation of productivity if total lipid mg/ L were added to the figure so that rates, concentration, and totals could be compared.

Author Response

Attachmented

Reviewer 2 Report

Article quite interesting paper.

However, the article need some slightly improvements in english language and, the scientific considerations (treatment of results and conclusions sections) can be improved.

Author Response

Thanks for providing a positive feedback on the manuscript. We considered your comment in the revised version.

Reviewer 3 Report

In the paper presented by Touliabah and Almutairi the authors investigated combined effect of N-depletion and 3-Indoleacetic acid (IAA) supplementation on lipid accumulation of the marine eustigmatophyte Nannochloropsis oceanica. Obtained results showed that addition of IAA enhanced both growth and lipid accumulation due to enhancement of pigments biosynthesis.  Although the results are explained really good combining literature data and results from this research, there is one major drawback of the manuscript -  the organization and result presentation. In this form it is hard to fallow the results since all the figures and tables are grouped together at the end of the manuscript. Because of that it would be good to make figures smaller and insert them in the manuscript where they are mentioned for the first time. Tables can retain their existing form but should be also be inserted in a proper place in the manuscript.

Some minor mistakes:

  1. Nannochloropsis oceanica is mentioned in this form trough the manuscript and in some places in oceanica   form. Make it uniform.
  2. row 69. 19.0 mg/Ld, zero should be removed
  3. row 77. Isochrysis sp. should be written with capital letter.
  4. row 94. F/2 should be f/2
  5. equation 5. A bracket should be removed
  6. row 166. Double %%. One should be removed.
  7. row 280. oceanica should be written in italic.

Round 2

Reviewer 3 Report

After the requested changes have been made, I consider that the work in the new form is acceptable for publication.